# The Effectiveness of Nutritional Interventions Implemented through Lady Health Workers on the Reduction of Stunting in Children under 5 in Pakistan: The Difference-in-Difference Analysis

**DOI:** 10.3390/nu16132149

**Published:** 2024-07-05

**Authors:** Khizar Ashraf, Tanvir M. Huda, Javeria Ikram, Shabina Ariff, Muhammad Sajid, Gul Nawaz Khan, Muhammad Umer, Imran Ahmed, Michael J. Dibley, Sajid Bashir Soofi

**Affiliations:** 1Sydney School of Public Health, Faculty of Medicine and Health, The University of Sydney, Sydney, NSW 2006, Australia; khizarashraf@gmail.com (K.A.); michael.dibley@sydney.edu.au (M.J.D.); 2South Western Sydney Local Health District, Liverpool, NSW 2170, Australia; 3The Tweed Valley Hospital, Cudgen, NSW 2487, Australia; javeriakhizar@gmail.com; 4Department of Pediatrics & Child Health, The Aga Khan University, Karachi 74800, Pakistan; shabina.ariff@aku.edu; 5Center of Excellence in Women & Child Health, The Aga Khan University, Karachi 74800, Pakistanimran.ahmed@aku.edu (I.A.)

**Keywords:** nutrition, Pakistan, stunting prevention, lipid-based nutrient supplements, nutritional interventions, programmatic review

## Abstract

In Pakistan, the 2018 National Nutrition Survey reported that 40% of children under five years old were stunted. This study assessed the effectiveness of nutritional supplementation in reducing stunting among children under five years old in two rural districts in Sindh, Pakistan. This was a mixed-method quasi-experimental study comprising intervention and control populations, with 3397 and 3277 children under five years old participating in the baseline and end-line surveys, respectively. The study areas were similar in terms of demographic and economic circumstances. In the intervention group, pregnant and lactating women (first six months post-partum) received wheat soy blend, children 6–23 months old received Wawamum (lipid-based supplement), and children 24–59 months old received micronutrient powders, all through lady health workers. This was underpinned by nutrition behaviour change communication for appropriate complementary feeding practices and hygiene promotion targeted at primary caregivers. The control group received no intervention. The impact was assessed using the difference-in-difference analysis with kernel propensity score matching to adjust the differences among the control and intervention populations. The overall DID analysis indicated that the intervention did not significantly reduce the prevalence of stunting (under 5 years) [DID = −5.1, *p* = 0.079]. The adjusted DID indicated a significant decrease of 13% [DID = −13.0, *p* = 0.001] in the number of stunted children 24–59 months of age at the endline survey. A significant reduction in underweight among children 24–59 months old was also observed (DID = −9.4%, *p* = 0.014). In conclusion, this evidence further establishes that nutrient uptake through an intervention for a short duration cannot effectively reduce stunting. It requires continuous nutritional supplementation for mothers during the pregnancy and an initial six months of lactation and then nutritional supplementation for children 6–59 months of age underpinned by effective behaviour change communication targeting mothers and other caregivers for improving complementary feeding practices and hygiene promotion.

## 1. Introduction

Undernutrition, seen mostly in low- and middle-income countries, is a key risk factor for death in children under five years of age [1]. Stunting is a marker of a deficient environment [2] and is associated with delayed child development, reduced earnings in adulthood, and chronic disease. Globally, 21.9% of children under five years are stunted, and 7.3% are wasted [3].

Before the onset of the Scaling Up Nutrition (SUN) movement, the long-term consequences of all forms of malnutrition gained the attention of researchers and practitioners globally, mainly after the publication of the Lancet series of 2008. The follow-up 2013 Lancet series extended the discussions to adolescent nutrition. The Sustainable Development Goals brought stunting to focus when targets changed from underweight to stunting. Since then, donor-funded programming has identified stunting reduction as a primary target; adequate linear growth will reduce the associated higher mortality and minimise the long-term impact on chronic conditions [4,5,6,7].

The nutritional status of children in Pakistan is alarming, with 40.2% of children under five years of age stunted, 17.7% wasted, and 28.9% underweight, with a higher prevalence of stunting in rural areas than in urban areas [3]. In Sindh province, 45.5% of children under five are stunted [3]. Food insecurity, poor health care, inadequate diets, lack of affordability of a balanced diet, inaccessibility to basic nutrients, gender and socio-demographic inequality, and lack of family planning are some of the core social and biological stressors contributing to the poor health of women and children in Pakistan [8]. Despite being a predominantly agrarian country, Pakistan is 11th among 148 countries on the Food Security Risk Index (FSRI), which highlights the population’s poor nutritional and health status [9]. Diets low in animal protein sources, fruits, and vegetables and high in phytates can contribute to micronutrient deficiencies, resulting in anaemia, developmental delays, and poor child growth [10,11]. Phytic acid is reported to inhibit iron, zinc, calcium, manganese, and magnesium absorption [12]. According to a recent survey, more than half (53.7%) of Pakistani children under five years of age are anaemic, with 5.7% being severely anaemic [3]. This indicates a deficient environment, possibly because of inadequate intake of food and nutrients or excess of illnesses. A deficient environment can delay child development and future work capacity [2].

The first 1000 days of a child’s life, from conception to two years, are the most vulnerable and critical for building the foundations of optimum child growth and development [13]. Studies aiming to reduce malnutrition in children by targeting the first 1000 days have shown improvements in birth and growth outcomes [14].

Common food-based interventions, including the provision of specialised nutritious food (SNF) during pregnancy, the first six months of lactation, and for children up to two years of age, have proven to be more effective and holistic in targeting nutrient deficiencies in children [10,15,16,17].

In low- and middle-income and food-insecure settings, complementary foods may include lipid-based nutrient supplements (LNS) and fortified blended foods (FBFs). FBFs are made from readily available grains and fortified to provide sufficient energy and nutrient density to prevent undernutrition [10]. The World Food Programme (WFP) developed a wheat soy blend (WSB), a mixture of protein, sugar, oil, soybeans, and adjusted micronutrients (Table 1) [18]. The similar products can improve birth outcomes, if given to pregnant women [19]. LNS is an energy-dense product containing micronutrients and essential fatty acids, which children can easily consume [20,21]. When provided as a complementary food for children, LNS improves nutrition-related outcomes in children [21]. The iLiNS-DYAD study found that children randomised to take LNS had greater length, length-for-age z-scores (LAZ), weight, weight-for-age z-scores (WAZ), and a reduced prevalence of stunting at 18 months compared to the other groups [8,15,22].

The primary objective of this study was to assess the effectiveness of nutritional supplementation in reducing stunting prevalence in children aged up to 59 months. The secondary objectives included reducing wasting and underweight in children up to 59 months of age.

## 2. Materials and Methods

### 2.1. Study Design

This was a mixed-method study [23]. The impact evaluation design was quasi-experimental, with one program and one comparison population, and both areas matched for demographic, ethnic, and economic circumstances [23,24]. The impact of the intervention was measured using baseline (2014) and endline (2018) cross-sectional survey data of children 6–59 months of age.

The intervention was implemented during 2014–2017 (4 years) in 29 union councils (UCs), wherein the study was conducted between August 2014 and September 2018 in the Thatta and Sajawal districts of the Sindh province.

Around 15–20 lady health workers (LHW) are attached to each public-sector-financed health centre in each UC. Each LHW provides health services to on average 90–100 households, making them the first and most integral part of Pakistan’s efforts to improve maternal and child health. Thus, we engaged the LHWs to implement this nutritional intervention in the intervention group while the control group received no interventions. The primary outcome of interest was the changes in the prevalence of stunting in children aged 6–59 months in the program compared with the control after four years of implementation. The secondary outcomes were the prevalence of wasting and underweight. Providing nutritional supplementation was impossible without behaviour change communication (BCC), emphasizing the importance of IYCF practices and hygiene promotion. The lady health workers (LHWs) were used as vehicles/channels for service delivery, which included the supply of supplements, nutrition BCC, and hygiene promotion, along with their routine health promotion activities in the health system.

The Ethics Review Committee (ERC) of Aga Khan University (AKU) and the National Bioethics Committee (NBC) of Pakistan granted ethical approval for the study, including human subjects. The AKU ethical review approval number is 2919-Ped-ERC-14, received on 31 January 2014. All included study participants provided written informed consent before enrolment. The study was a quasi-experiment registered on ClinicalTrials.gov with ID: NCT02422953.

### 2.2. Study Setting and Participants

The study districts (Thatta and Sajawal) are administratively comprised of 55 union councils (UCs) and nine talukas with a population of approximately 1.8 million, and each UC has at least one primary healthcare centre, operated by the public sector. We selected 29 of these UCs/talukas which had LHW coverage as the intervention areas. Of the 29 UCs/talukas, the areas without LHWs coverage were selected as the control areas. All children 6–59 months of age were eligible for enrolment in the study in both the intervention and control areas (Figure 1).

In the intervention group, pregnant and lactating women received wheat soy blend (WSB), children 6–23 months of age received a lipid-based nutrient supplement (LNS) branded as Wawamum, and children 24–59 months were given a multi-micronutrient powder (MNP). This supply of nutritional supplements was underpinned by behaviour change communication for improving complementary feeding practices, as well as hand hygiene promotion targeted to their primary caregivers (mostly mothers). The control group received routine healthcare and none of the above-mentioned nutritional interventions.

### 2.3. Selection of Study Areas

Thatta and Sajawal districts were selected based on their high prevalence of stunting and all forms of malnutrition. In addition, there was a high coverage of lady health workers (LHW) in parts of the district, and there was the possibility of future scaling up. All the intervention areas were rural and were similar to other underprivileged areas of Sindh and Pakistan where similar programs can be anticipated. We implemented the intervention in parts of UCs with LHW coverage.

### 2.4. Data Collection

The study data were collected at the beginning of the intervention (2014) and towards the end (2017). We used a standard pre- and post-survey questionnaire. The data collection teams comprised four female data collectors and a male team leader, who were hired locally. The data collectors had a minimum high school and the team leaders had graduate-level education. All teams received six-day training on anthropometric measurements, ethical considerations, and data collection techniques. Before leaving for the field survey, they had a one day of practical training session in the field. Team leaders carried with them a study manual. Field-tested questionnaires were used for data collection. The baseline and endline questionnaires covered sociodemographic characteristics, health-seeking behaviour, past intervention exposure, and anthropometric data. The data collectors used printed copies of the questionnaires in the field.

### 2.5. Procedures

The LHWs distributed all the nutritional supplements to the intervention group. The pregnant and lactating women (PLW) in the intervention group were given wheat soy blend (WSB). The WSB consisted of partially cooked wheat and soy fortified with micronutrients (Table 1). Each PLW received 5 kg (165 g per day) of WSB every month for the duration of pregnancy and through the first six months of lactation. Children 6–23 months of age were given LNS branded as Wawamum (Table 1). Children 24–59 months received multiple micronutrient powders (MNP) (Table 1). PLW and children 6–59 months in the control group received standard routine care. In both the intervention and control group, the nutritional status of children 6–59 months, including stunting, wasting, and underweight, was assessed.

LHWs also performed a monthly follow-up during program implementation to assess adherence to the intervention. We used participant recall and observations of used and unused LNS sachets and WSB to collect data on adherence at all intervention households at each visit. Anthropometry was performed by trained survey staff using digital scales (Seca 874, Seca, Hamburg, Germany), height boards (Seca 213, Seca Germany), and infant meters (Seca 210, Seca Germany). The Agha Khan University team trained the survey staff, which included standardisation exercises during the initial orientation before the baseline and end-line surveys.

A monitoring team conducted the quality assurance of the data. The team randomly visited 5% of the households to validate data collection. Team leaders supervised the data collection teams, and a field supervisor and project manager conducted further data monitoring. The field supervisor ensured data quality through ENA-SMART software version 2011 by conducting weekly plausibility checks for anthropometric measurements.

### 2.6. Sample Size

The sample size for this study was determined to compare two cross-sectional surveys to assess the impact of intervention in achieving a 10% reduction in the prevalence of stunting over the four-year implementation period. We based our calculations on a baseline stunting prevalence of 49% in Sindh to estimate the potential impact of the interventions (with α = 0.05 and power = 0.80). The significance level was set at *p* < 0.05. Each of the baseline and endline groups should have consisted of 3200 participants. Within the 29 UCs/talukas of Thatta and Sajawal, all households with children 6–59 months of age were eligible for the baseline and endline surveys. The eligible participants were randomly selected from LHW family household list.

### 2.7. Statistical Analysis

Data were entered into the Visual FoxPro database and analysed using STATA version 17 (Stata/SE 17 Stata Corporation, College Station, TX, USA). Household characteristic data were analysed for all participants who completed the survey. The wealth quintiles were constructed using principal component factors (pcf) for data reduction and extraction of the maximum variance [7]. This includes individual, household, and community access to resources. The composite score included 33 variables: ownership of assets, land, livestock, household construction, and access to sanitation facilities (Appendix A).

We used DID analyses for repeated cross-sectional surveys (baseline and end line) to examine the program’s impact on child outcomes by utilising the value of the propensity score [24,25,26]. The DID relies on two differences. The first difference was across time periods separately between the intervention and control groups and the second difference was the difference obtained in the first step between the intervention and control groups [26]. A difference-in-differences test among matched subjects estimated the mean change in outcomes in the intervention group after subtracting the mean change in outcomes in the control group.
DID = (Intervention_endline_ − Intervention_baseline_) − (Control_endline_ − Conrol_baseline_) 

The DID was obtained using the regression approach taking the interaction of group × time.
Y = β0 + β1 × [Time] + β2 × [Intervention] + β3 × [Time × Intervention] + β4 × [Covariates] + ε 

Propensity score matching constructs a statistical comparison group based on a model of the probability of participating in treatment using the observed characteristics [25]. We then matched the participants to non-participants based on the probability or propensity score. Variable selection for matching was based on theoretical relation to anthropometric outcomes and included maternal education, wealth quintiles, hand hygiene indicators, gender, household density, and food insecurity.

## 3. Results

We analysed a total of 3397 children 6–59 months (1832 intervention, 1565 control) at baseline compared to 3277 children (1650 intervention, 1627 control) at the end-line survey (Table 2).

### 3.1. Baseline Characteristics

At baseline, the average household density of the intervention group was 7.1 compared to 6.2 in the control group (Table 2). There was a statistically significant difference in the chi-squared test between the intervention and control groups (*p* = 0.00). Among the children aged 6–59 months, 51.9% of males and 48.1% of females were analysed in the intervention group compared to 51.1% of males and 48.9% of females in the control group.

### 3.2. End Line Characteristics

At the end line, the average household density in both the intervention and control groups was 7.3. There was no statistically significant difference in the chi-squared test between the intervention and control groups (*p* = 0.689). Among the screened children aged 6–59 months, 48.9% were males and 51.1% females in the intervention group, compared to 53.3% males and 46.7% females in the control group. In the intervention group, 82.9% of the mothers had no education compared to 93.2% in the control group. In total, 4.6% and 12.5% of the mothers in the intervention group had secondary or higher education and primary or middle education, respectively.

### 3.3. Effects of the Nutritional Supplementation Program on the Prevalence of Stunting

Table 3 presents the results from the difference-in-differences analysis, illustrating the differential changes in childhood stunting and wasting rates between the intervention and control arms. Overall, the intervention had a large differential change in childhood stunting, but it was not statistically significant [DID = −5.1 percentage points (pp), *p* = 0.079]. However, for children aged 24–59 months, the study found that the intervention had a more pronounced and statistically significant effect on stunting, with a difference in stunting prevalence of −13 percentage points [*p* = 0.001]. The intervention showed a stronger effect on girls (DID = −5.7 pp, *p* = 0.171) than boys (DID = −3.9 pp, *p* = 0.334), although neither was significant. We also assessed the impact of the intervention across various wealth groups. We observed a substantial variation in the intervention’s effect on childhood stunting among different wealth groups. The difference in stunting prevalence was most pronounced, with a decline of −21.2 percentage points [*p* = 0.001] in the highest wealth quintile.

### 3.4. Effects of the Nutritional Supplementation Program on the Prevalence of Wasting

As presented in Table 3, the intervention had no significant impact on wasting [DID = −2.1 percentage points (pp), *p* = 0.298]. The intervention showed a more substantial effect on boys, with a difference in the prevalence of wasting at −6.4 percentage points [p = 0.027]. In contrast, there was no effect for the girls’ group (DID = 3.3 pp, *p* = 0.239). We also assessed the impact of the intervention across various wealth groups. We observed a substantial variation in the intervention’s effect on childhood wasting among different wealth groups. The difference in wasting prevalence was most pronounced, with a decline of −11 percentage points [*p* = 0.037], in the second (lowest) wealth quintile.

### 3.5. Effects of the Nutritional Supplementation Program on the Prevalence of Underweight

As presented in Table 3, overall, the intervention had a large statistically significant impact on underweight [DID = −8.2 pp, *p* = 0.004]. For children aged 24–59 months, the study found that the intervention had a more pronounced and statistically significant effect on underweight, with a difference in the prevalence of underweight of −9.4 pp [*p* = 0.014]. We also assessed the impact of the intervention across various wealth groups. We observed a substantial variation in the intervention’s effect on childhood underweight among different wealth groups. The difference in underweight prevalence was most pronounced, with a decline of −26.9 percentage points [*p* = 0], in the highest and second lowest (DID = −14.7, *p* = 0.031) wealth quintiles.

## 4. Discussion

Although the primary outcome was not achieved, the intervention still demonstrated a significant reduction in underweight among children aged 6–59 months, with a substantial impact on childhood stunting, particularly in older age groups and higher wealth quintiles. However, no significant impact was observed on wasting. Notably, while the intervention significantly reduced stunting in children aged 24–59 months, this effect was not observed in younger children aged 6–23 months, though a large differential change in stunting was observed.

Stunting and underweight, though easily measurable in surveys, may not fully capture overall health status, nutritional deficiencies, or child development. Stunting often reflects deficient environments influenced by nutrient and food limitations [2]. Nutritional supplementation may have played a significant role in improving the deficient environment, leading to reduced stunting and underweight.

An RCT conducted on a subset of our study population showed promising evidence [8,23] that LNS during 6–23 months of age improved linear growth and reduced stunting in children at 24 months of age [27]. When the intervention is implemented in the controlled circumstances of a randomised controlled trial, it impacts child growth. However, the impact is less clear when the intervention is delivered in a public sector program. We observed that the RCT focused on 6–23-month-old children, whereas the quasi-experimental study included children up to 59 months in the intervention. The reasons for this difference in outcome between the RCT and our study could be manifold, including uninterrupted supply chain of LNS, more engagement and prioritisation by lady health workers, more compliance and non-sharing of LNS with other siblings, and effective behavioural change communication (BCC). In the quasi-experimental study, the challenge becomes more when BCC is merely replaced by education or awareness sessions that do not focus on small doable actions which are enablers of a behaviour change [28,29]. The lack of an impact in the younger age group might be shorter exposure to nutrient supplements compared to older children. In our study, these older children between 24 and 59 months would have completed their LNS up to 23 months of age and continued with MNP, but the younger age group was still receiving the supplements. It is difficult to ascertain strict adherence to the daily consumption of LNS. There are also chances for an interrupted supply chain for LNS in the far-flung areas of Thatta and Sajawal, which could be a reason for observing the impact in the highest wealth quintiles and older age groups as they had more access to other foods and more nutrients, and the supplement just helped them to bridge the minimum requirement. Supply chain delays can easily be considered as reasons for the unmet needs of children from less wealthy households and younger age groups. It can also be argued that since the impact of LNS supplementation in an RCT [8] performed on a subset of these children was only observed at 24 months, this could be considered as a minimum supplementation period for observing any significant impact. It also questions the effectiveness of maternal supplementation alone if the impact is only visible after the child has completed supplementation up to 24 months of age. In addition, the highest wealth quintile and older age groups (longer supplementation) clearly show a significant DID in stunting levels, illustrating the impact of nutritional supplementation in improving one element of the deficient environment, but this may not have been the case for children from lower wealth quintiles and younger age groups, and thus is a non-significant DID result. Another finding is that the improvement was greater in boys than in girls. A lower impact on girls can be attributed to gender preference, which is well-documented in the local context [30,31].

Studies also suggest that combining BCC with supplementary foods in deficient or food-insecure environments improves dietary diversity [25]. Our interventions addressed the nutrient gap through supplementation underpinned by behaviour change communication and was delivered through a public sector mechanism. This showed that such an intervention is possible and sustainable if the local supply chain is established. However, it is possible that the complimentary feeding at homes is substantially inadequate and the supplementation through nutrition programs may not be sufficient to bridge the nutrient gap [25]. We have also observed that the change in stunting reduction is significant in high-wealth quintiles, which could be because of a lesser nutrient gap, which was bridged by the supplementation and underpinned by BCC. The significant impact on underweight and differential change in stunting supports that further scale up may be explored. The impact of this intervention will have an intergenerational impact in the lives of many in improving their deficient environment. This will not only reduce the burden on health system, but it will also reduce the deficient environments in which stunting is an outcome and leads to benefits beyond the health sector. Given that it was the first such effort in Pakistan, it is critical that scale-up is considered with designer thinking and locally acclimatised to each of the focus districts.

Our trial presented challenges similar to many other trials where short- to medium-term assessments of stunting prevention programs (interventions) have not significantly reduced stunting [25]. These programs may have achieved other outcomes, but their limited impact on stunting may undermine their credibility in front of policymakers [25]. Evidence suggests that the reduction in stunting prevalence also depends on underlying factors such as the social and political situation, economic factors, and sanitation conditions [32].

Our study population had levels and patterns of child undernutrition comparable to many other low- and middle-income countries. Studies have found that the prevalence of stunting and underweight gradually increases with age from birth until 24 to 36 months [33]. We found similar patterns in our data where stunting and underweight were higher with increasing child age, especially in children older than two years [6,34]. Our findings are similar to those of studies conducted in other Asian countries [34,35]. Also, a study from Nigeria suggested that appropriate supplementation in the first 1000 days of life can avert stunting [36]. Studies also observed that stunting levels reduce as household wealth increases, a finding we observed in our study [6].

This study had a large sample, with matching controls within the district, and documented good participation. This study used DID analysis and the propensity score matching approach to estimate impact. It also allows for the evaluation of an LNS-based product approach underpinned by BCC implemented through lady health workers (LHW). It evaluated programs in the public sector where differences (including the presence of LHW) existed between the program and comparison districts at baseline and the end line. We used a cross-sectional survey design with intervention and control areas at baseline and end-line surveys to assess the absolute effect of the intervention program. However, the stunting prevention program has some limitations. First, we did not have data on exposure to LNS and MNP for children aged 6–59 months. As cross-sectional surveys cannot establish causality, we could not link the reduction in stunting prevalence to the consumption of LNS and MNP. Second, the impact on children 6–23 months could be due to the exposure of their mothers to supplements during pregnancy and lactation, as well as the children’s own exposure to LNS. Hence, exposure to both supplements may have contributed to the reduction in stunting, wasting, and underweight among older children 24–59 months at the end line. In addition, occasional and continued exposure to other donor-funded projects in the study areas is possible.

It is important to consider that the real goal of nutritional programs is not only to change the statistical indicators but also to reduce the long-term consequences and child mortality linked to poor linear growth. This change will require social and political commitments leading to multi-sectoral interventions that will improve deficient environments, including targeting the affordability of families to access essential nutrients. Standalone interventions will not bring drastic changes across the population. This study clearly shows the need to look at indicators beyond stunting, wasting, and underweight to assess program performance. These could include but are not limited to anaemia in children and pregnant mothers, as well as change in the infant and young child feeding practices, thus improving dietary diversity and minimum acceptable diet. From a policy perspective, the gains documented in this study show that nutritional supplementation underpinned by effective BCC may have contributed to improving the deficient environment. However, we need to do more research to identify the real nutrient gap. This will help improve the deficient environments and make a difference in the lives of those constrained by the limited access and availability of nutritious food. The researcher and policymakers may also consider options for better connectivity with the program participants by deploying technological solutions (like mobile apps) to engage and improve the BCC.

## 5. Conclusions

In conclusion, nutritional supplementation and behaviour change communication targeting pregnant and lactating women (6 months post-partum) and children 6–23 months of age (the first 1000 days) reduced stunting and underweight among children aged 24–59 months. This further establishes that nutrient uptake through an intervention for a short duration cannot effectively reduce stunting and underweight. It requires continuous nutritional supplementation for mothers during pregnancy and the initial six months of lactation and then nutritional supplementation for children 6–59 months of age underpinned by effective behaviour change communication targeting mothers and other caregivers for improving complementary feeding practices and hygiene promotion. It can be an effective and scalable intervention for improving the nutritional status of children.

## Figures and Tables

**Figure 1 nutrients-16-02149-f001:**
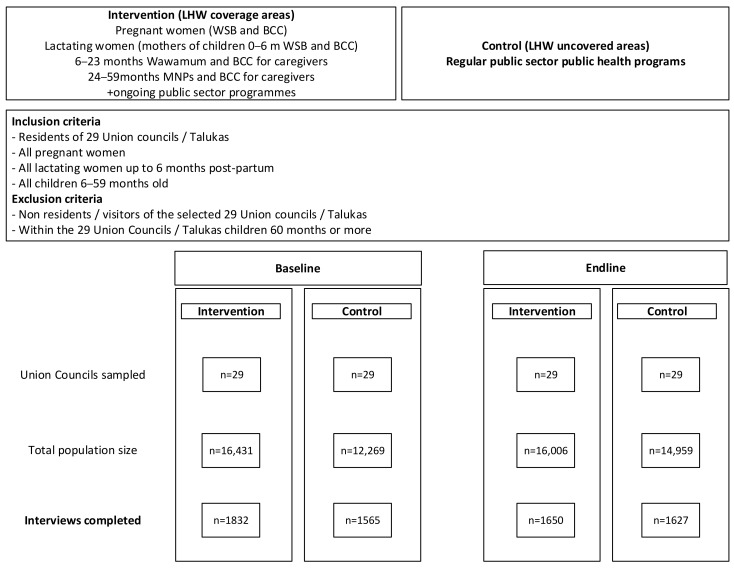
Baseline and endline study participation in intervention and control groups in Thatta and Sajawal districts in 2014–2018.

**Table 1 nutrients-16-02149-t001:** Contents of nutritional supplements.

Ration Contents	WSB	Wawamum (per 50 G)	MNP (per 1 G)
Daily ration (g/person/day)	167	50	On alternate day
Energy (kcal)	633	255	-
Protein (g)	29.1	5.5	-
Fat (g)	10.2	13	-
Calcium (mg)	683	267.5	-
Iron (mg)	13.9	5	10
Iodine (µg)	67	50	90
Vitamin A (µg RE)	842	275	400
Thiamine B1 (mg)	0.66	0.5	-
Riboflavin vitamin B2 (mg)	1.03	1.05	-
Niacin (mg NE)	15.3	6.5	6
Vitamin C (mg)	168.9	30	30
Pantothenic VitB5 (mg)	3.4	2	-
Vitamin B6 (mg)	1.8	0.9	0.5
Vitamin B7 (µg)	-	30	-
Folic Acid (µg)	100	-	902
Vitamin B12 (µg)	3	1.35	0.9
Vitamin D (µg)	10.0	7.5	-
Vitamin E (mg)	-	8	-
Vitamin K (µg)	-	13.5	-
Ca (mg)	-	-	-
Cu (mg)	-	0.7	-
Magnesium (mg)	-	75	-
Manganese (mg)	-	0.6	-
Phosphorus (mg)	-	225	-
Potassium (mg)	-	450	-
Selenium (µg)	49.3	10	17
Na (mg)	-	135	-
Zn (mg)	-	5.5	-
Vitamin D3 (μg)	-	-	5
Vitamin E (mg)	15.8	-	5
Vitamin K1 (µg)	-	-	-
Vitamin B1 (mg)	-	-	0.5
Vitamin B2 (mg)	-	-	0.5
Folic acid (µg)	-	-	-
Zinc (mg)	11.2	-	4.1
Copper (mg)	0.6	-	0.56
Folate (µg)	288	165	-
Dry skimmed milk protein	-	1.8	-
ω-3 fatty acids	-	0.15	-
ω-6 fatty acid	-	1.3	-

**Table 2 nutrients-16-02149-t002:** Child and household characteristics.

Characteristics	Baseline					Endline				
	Intervention		Control		*p*	Intervention		Control		*p*
	n = 1832	%	n = 1565	%		n = 1650	%	n = 1627	%	
Average household density	7.1		6.2		0.00	7.3		7.3		0.356
**Child sex**					0.668					0.012
Male	950	51.9%	800	51.1%		807	48.9%	867	53.3%	
Female	882	48.1%	765	48.9%		843	51.1%	760	46.7%	
**Age**					0.011					0.508
6–23 months	849	46.3%	657	42.0%		740	44.8%	711	43.7%	
24–59 months	983	53.7%	908	58.0%		910	55.2%	916	56.3%	
**Hand washing access ***	1549	84.6%	1285	82.1%	0.00	877	53.2%	760	46.7%	0.00
**Maternal education**					0.00					0.00
None	1469	80.2%	1536	98.1%		1368	82.9%	1517	93.2%	
Primary or middle	267	14.6%	27	1.7%		206	12.5%	90	5.5%	
Secondary or higher	96	5.2%	2	0.1%		76	4.6%	20	1.2%	
**Wealth quintiles**					0.00					0.00
Lowest	187	10.2%	509	32.5%		223	13.5%	449	27.6%	
Second	266	14.5%	405	25.9%		262	15.9%	397	24.4%	
Middle	353	19.3%	326	20.8%		321	19.5%	343	21.1%	
Fourth	446	24.3%	251	16.0%		402	24.4%	251	15.4%	
Highest	580	31.7%	74	4.7%		442	26.8%	187	11.5%	

* Handwashing access includes soap, cleaning agent, and water availability at the household.

**Table 3 nutrients-16-02149-t003:** Effects of the nutritional supplementation program on the prevalence of stunting, wasting, and undernutrition **, and difference-in-difference estimation.

	Baseline				Endline					
	Intervention	Control	Difference	*p*	Intervention	Control	Difference	*p*	DID	*p*
**Stunting** **(overall)**	55.7	55.2	0.4	0.849	44.6	49.3	−4.7	0.01	−5.1	0.079
**Gender**										
Boys	56.4	58.5	−2.2	0.485	45.2	51.4	−6.1	0.018	−3.9	0.334
Girls	54.7	52.5	2.3	0.489	44	47.4	−3.5	0.183	−5.7	0.171
**Age Group**										
6–23 m	51.5	54.9	−3.4	0.307	39.7	39.2	0.5	0.844	4	0.358
24–59 m	58.9	55	3.9	0.196	48.5	57.7	−9.1	0.00	−13	0.001
**Wealth Quintiles**										
Lowest	53.6	50.2	3.5	0.477	47.3	48.2	−0.9	0.853	−4.4	0.533
Second	58.4	57.5	0.9	0.844	50.6	50.2	0.4	0.934	−0.5	0.94
Middle	59.8	62	−2.2	0.639	48	49.4	−1.5	0.738	0.7	0.908
Fourth	57.5	58.9	−1.3	0.797	45.8	51.5	−5.2	0.174	−3.9	0.547
Highest	46.5	36.7	9.8	0.079	36.2	47.6	−11.4	0.001	−21.2	0.001
**Wasting (overall)**	17.7	16.2	1.6	0.314	11.7	12.2	−0.5	0.679	−2.1	0.298
**Gender**										
Boys	19.6	14.4	5.5	0.013	12.8	13.6	−0.9	0.636	−6.4	0.027
Girls	15.5	18.9	−3.4	0.122	10.6	10.7	−0.1	0.958	3.3	0.239
**Age Group**										
6–23 m	25.6	16.2	9.4	0.00	18.1	14	4.1	0.052	−5.3	0.114
24–59 m	10.8	15.9	−5.1	0.005	6.4	10.6	−4.2	0.004	0.9	0.712
**Wealth Quintiles**										
Lowest	16.7	21.1	−4.4	0.206	11.7	13.5	−1.8	0.616	2.5	0.613
Second	23.8	18.2	5.5	0.123	14	19.5	−5.5	0.156	−11	0.037
Middle	18.6	23.7	−5	0.148	11.3	13.5	−2.2	0.497	2.8	0.552
Fourth	17.2	13.5	3.7	0.302	13	12.6	0.5	0.862	−3.3	0.466
Highest	10.9	4.8	6.1	0.063	9.3	9.2	0.2	0.933	−5.9	0.12
**Underweight (overall)**	47.8	42.7	5.1	0.021	33.6	36.7	−3.1	0.078	−8.2	0.004
**Gender**										
Boys	48	43.9	4.1	0.184	35.4	38.2	−2.9	0.258	−6.9	0.081
Girls	47.5	42.9	4.6	0.146	31.8	35.1	−3.3	0.188	−7.9	0.051
**Age Group**										
6–23 m	46.5	36.2	10.4	0.001	36.5	29.4	7.2	0.006	−3.2	0.444
24–59 m	48.4	50.6	−2.2	0.453	31.2	42.7	−11.6	0.00	−9.4	0.014
**Wealth Quintiles**										
Lowest	46.8	46.6	0.2	0.973	33.3	38.7	−5.3	0.285	−5.5	0.424
Second	50.9	48.2	2.7	0.566	34.9	47	−12.1	0.015	−14.7	0.031
Middle	53.5	50.3	3.2	0.493	39.4	36.9	2.5	0.575	−0.8	0.906
Fourth	45.5	45.2	0.3	0.961	36.3	39.6	−3.3	0.381	−3.6	0.578
Highest	41.7	20.2	21.5	0	26.2	31.6	−5.4	0.08	−26.9	0

** Wasting is defined as low weight-for-height. Stunting is defined as low height-for-age. Underweight is defined as low weight-for-age.

## Data Availability

Restrictions apply to the availability of these data. Data are available on request with the permission of the corresponding author of Aga Khan University, Pakistan.

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
