# Peer review of "The Effectiveness of Nutritional Interventions Implemented through Lady Health Workers on the Reduction of Stunting in Children under 5 in Pakistan: The Difference-in-Difference Analysis"

_nutrients, 2024, doi:10.3390/nu16132149_

Round 1

Reviewer 1 Report

Comments and Suggestions for Authors

Overall, this report is difficult to understand with many confusing statements. Key issues are:

The study design is unclear. Was this a cluster-randomised trial? If so, this needs to be reflected in the sample size calculation.

Page 2: “The primary objective of this study was to assess the effectiveness of nutritional supplementation in reducing stunting prevalence in children aged up to 59 months.” However, the study design is not appropriate for this aim as the intervention comprised of “behaviour change communication for proper complementary and feeding practices, and hand hygiene targeted to their primary caregivers (mostly mothers)” and the input of LHWs as well as nutritional supplements. And what about MNPs? Only the effects of the combined intervention can be assessed.

Was the study registered in a trials database?

Other issues:

The abstract does not capture the main elements of the study.

It is not clear how the intervention and control groups were created and what inputs the two groups received. Clarify the duration of the intervention. State the ages of the children who were recruited.

“The participants received nutrient supplements underpinned by nutrition behaviour change communication for appropriate complementary feeding practices and hand hygiene targeted at primary caregivers during the first 1000 days of an infant's life”. Clarify that nutritional supplements were given to mothers during pregnancy and up to six months post-partum and children between 6–24 months of age.

“first 1000 days of an infant's life”: first 1000 days extends before and beyond infancy – suggest replace “infant” with “child”.

What does “BCC” refer to?

The statement “It necessitates continuous supplementation for 1000 days and beyond, supported by social, economic, and political support.” is not based on the findings of this study.

Page 2

The section “According to a recent survey, more than half (53.7%) of Pakistani children under five years of age are anaemic, with 5.7% being severely anaemic [8]. This indicates a deficient environment, possibly be- cause of inadequate intake of food and nutrients or excess of illnesses. A deficient environment can delay child development and future work capacity[2].” Could be moved to the previous paragraph.

Page 3, 2nd para: delete “that would prevent stunting, wasting, and micronutrient deficiencies,” as the study aims to determine whether or not this occurred.

The statement in the abstract “The overall DID analysis indicated that the intervention did not significantly reduce the prevalence of stunting (under-5-years) [DID = −5.1, p 0.079]“ is not compatible with “The primary outcome of interest was the changes in the prevalence of stunting in children aged 6–23 months ..” on page 3. Please clarify what the primary outcome of the study.

 Page 4: The two sentences “Children 24-59 months received multiple micronutrient powders (MNP) (Table 1). This intervention occurred during the first 1000 days of an infant's life.” seem to be conflicting.

The sample size calculation is not appropriate for a cluster-randomised trial and does not state the desired sample size.

“The composite score included 33 variables: ownership of assets, land, livestock, household construction, and access to sanitation facilities.” Why are only 5 variables listed?

Page 5: “At baseline, the household density of the intervention group was 7.1 compared to 6.2 in the control group (Table 2).” Clarify if these values are mean or median and state either SD or IQR (including in the table).

Data on the clusters (names, number of children etc.) should be included in a supplementary file.

Comments on the Quality of English Language

There are some grammatical and syntax errors that need to be addressed.

Author Response

Thank you for your feedback.  Please see the attachment.

Reviewer 2 Report

Comments and Suggestions for Authors

Dear Authors,

The manuscript (nutrients-3002386) submitted for review is interesting.

Why did the authors decide to publish the results obtained in 2014 and 2017 only now.

 Authors, Please note and address the following comments:

Keywords:

I suggest adding "intervention" to key words

Introduction

The Introduction section is well written.

Material and Methods:

The material and methods are well written. Please provide the Ethical approval number of the Ethics Review Committee (ERC) of Aga Khan University (AKU) and the National Bioethics Committee (NBC) of Pakistan. Why did two institutions give ethical approval?

Please explain why this supplement was chosen and not another one in this experiment.

Results

In table 3, the second row is illegible. Maybe it's better to write I, II, III in the head of the table and explain it below the table. There is no need to write p-value either, you can just write p. Please correct this.

Limitation

I have a question. Is there any limitation to these results? If yes, it is worth writing about it. Authors should add a separate section of limitations to these results.

Conclusion

The current conclusions are well written.

References

References are not cited according to journal rules. Publications from MDPI provide information on how to properly cite. Authors may also find this information in the authors' guide.

 Despite my comments, I believe that it concerns an important area of research in an international context.

Author Response

(The authors gave the same response as above.)

Reviewer 3 Report

Comments and Suggestions for Authors

The manuscript is well written. However, there are some issues

Abstract: ok

Introduction: Regarding Phytates, their effect on mineral absorption should be stated.

Materials and Methods: the power of the site should be computed. Also, statistical significance.

Results: Figure 1 should be improved inclusion and exclusion criterias….

                                             Table 2 should be converted into a chart

                                             Table 3 should be redesigned

Discussion: Further discussion on DID should be added here or in the methods section ( as a suggestion remaid on Card and Krueger 1994)

Conclusions: OK

Author Response

(The authors gave the same response as above.)

Round 2

Reviewer 1 Report

Comments and Suggestions for Authors

The authors have addressed several of the issues that I raised previously well. These are clearly important data given the size of the trial, the combination of interventions tested and the vulnerability of the population.

However, I do think there are still some outstanding issues that need to be addressed.

Abstract

Clarify the identity of the two groups (intervention vs. control) or merge with the preceding sentence: “3397 and 3277 under-five children participated in the baseline and end-line surveys.”

Would “Both study areas were matched for demographic and economic circumstances.” be better as “The study areas were similar for demographic and economic circumstances.” to avoid confusion with the matching that occurred later in the analysis (Page 5) “Variable selection for matching was based on theoretical relation to anthropometric outcomes and included maternal education, wealth quintiles, hand hygiene indicators, gender, household density, and food insecurity.” and also because of the differences in these variables between intervention and controls (Table 2)?

“…of age at the survey.” Specify this refers to the end-line survey.

The statement “This evidence further establishes that nutrient uptake for a short duration cannot effectively reduce stunting and underweight.” Appears correct for stunting and is based on the primary outcome. However, “and underweight” should be deleted – as this was not the primary outcome and no data re underweight are presented in the abstract. Also, “nutrient uptake” was not measured in the study. This should be changed to “a combination of nutritional supplements during pregnancy, lactation and early life, behaviour change communication and hygiene promotion…” or something similar (i.e. the intervention that was tested).

The final sentence (“It requires continuous nutritional supplementation for mothers during the pregnancy and initial six months of lactation and then nutritional supplementation for children 6-59 months of age underpinned by effective behaviour change communication targeting mothers and other caregivers for improving complementary feeding practices and hygiene promotion.” Is not based on the finding of this study and may, or may not, be true. This would be better stated as “…continuous nutritional supplementation… should be tested in future research.”

Page 3:

“The primary objective of this study was to assess the effectiveness of nutritional supplementation in reducing stunting prevalence in children aged up to 59 months.” seems incorrect as a combined intervention was used. The study design prevents the evaluation of nutritional supplements alone. As a result, the sentence in the discussion “It also allows evaluating an LNS-based product approach implemented through Lady Health Workers (LHW).” appears misleading.

State “LHW” in full on first usage.

State if the WSB used was the WHO formula (what animal-sourced protein was included?).

Page 3: “the areas without LHWs coverage were selected as the control areas.” and Page 4 “The control area was a non- LHW area.” And page 5: “…whereas the remaining uncovered areas were classified as control.” The presence/absence of LHWs is an important difference between the two groups and additional to the intervention; it should be included in the discussion section. (This can just be stated once in the methods section).

Page 5:

“adherence to the intervention.” was assessed but no data are reported. This relates to the comment in the discussion section “It is difficult to ascertain strict adherence to the daily consumption of LNS.” and “This showed that such an intervention is possible and sustainable ..”and also “First, we did not have data on exposure to LNS and MNP for children aged 6-59 months.” It would be very useful to clarify what adherence data was available for the different groups and interventions. Did adherence differ between the interventions? Was poor adherence a likely cause of the limited impact of the interventions?

“each of the baseline and endline groups should have consisted of 3200 participants.” – this is presumably a total of approx. 6400 at each survey. Since this target was not reached, is the study underpowered to detect the desired effect of 10% reduction in stunting? If so, this should be included in the discussion. A statistically significant difference in stunting frequency between groups may have been observed if the sample size had been achieved.

Sections 3.1.1 and 3.1.2 could focus on the statistical differences between groups at baseline and end-line as presented in Table 2.

Discussion section

Page 10

The opening sentence “The intervention demonstrated a significant reduction in underweight among children aged 6-59 months, with a substantial impact on childhood stunting, particularly in older age groups and higher wealth quintiles.” is misleading as it does not reflect the lack of a statistically significant effect on the primary outcome.

“An RCT conducted on a subset of our study population …” was this a separate RCT or a subgroup analysis within the existing RCT? If a separate RCT, it would be useful to clarify if the intervention was LNS alone (without the other elements assessed in this study).

The lack of a statistically significant effect on the primary outcome does not support the statements “The significant impact on underweight and differential change in stunting support that scale up will benefit the community.” and “For policy perspective the gains documented in this study shows that food supplementation underpinned by effective BCC is critical for improving the deficient environment.”

“The impact may feel small on percentages…” is not consistent with section 3.1.1 “Overall, the intervention had a large differential change in childhood stunting,”.

Author Response

(The authors gave the same response as above.)

Reviewer 3 Report

Comments and Suggestions for Authors

The manuscript has been improved.

Author Response

Thank you very much for acknowledging that the manuscript has been improved.